# PIVEN: A Deep Neural Network for Prediction Intervals with Specific Value Prediction

## Abstract

Improving the robustness of neural nets in regression tasks is key to their application in multiple domains. Deep learning-based approaches aim to achieve this goal either by improving their prediction of specific values (i.e., point prediction), or by producing prediction intervals (PIs) that quantify uncertainty. We present PIVEN, a deep neural network for producing both a PI and a prediction of specific values. Unlike previous studies, PIVEN makes no assumptions regarding data distribution inside the PI, making its point prediction more effective for various real-world problems. Benchmark experiments show that our approach produces tighter uncertainty bounds than the current state-of-the-art approach for producing PIs, while maintaining comparable performance to the state-of-the-art approach for specific value-prediction. Additional evaluation on large image datasets further support our conclusions.

## 1 Introduction

Deep neural networks (DNNs) have been achieving state-of-the-art results in a large variety of complex problems. These include automated decision making and recommendation systems in the medical domain (Razzak et al., 2018), autonomous control of drones (Kaufmann et al., 2018) and self driving cars (Bojarski et al., 2016). In many of these domains, it is crucial not only that the prediction made by the DNN is accurate, but rather that its uncertainty is quantified. Quantifying uncertainty has many benefits, including risk reduction and more reliable planning (Khosravi et al., 2010).

In regression, uncertainty is quantified using prediction intervals (PIs), which offer upper and lower bounds on the value of a data point for a given probability (e.g., 95%). Existing non-Bayesian PI generation methods can be roughly divided into two groups: *a)* performing multiple runs of the regression problem, as in dropout (Gal & Ghahramani, 2016) or ensemble-based methods (Lakshminarayanan et al., 2017), then deriving post-hoc the PI from prediction variance, and; *b)* dedicated architectures for the PI generation (Pearce et al., 2018; Tagasovska & Lopez-Paz, 2019).

While effective, both approaches have shortcomings. On the one hand, the former group is not optimized for PIs generation, having to convert a set of sampled values into a distribution. This lack of PI optimization makes using these approaches difficult in domains such as financial risk mitigation or scheduling. For example, providing a PI for the number of days a machine can function without malfunctioning (e.g., 30-45 days with 99% certainty) is more valuable than a prediction for the specific time of failure. On the other hand, the latter group—PI-dedicated architectures —provides accurate upper and lower bounds for the prediction, but lacks in the accuracy of its value predictions. Consequently, these approaches select the middle of the interval as their value prediction, which is (as we later demonstrate) a sub-optimal strategy that makes assumptions regarding the value distribution within the interval. The shortcomings of PI-dedicated architectures with regard to value prediction are supported both by Pearce et al. (2018) and by our experiments in Section 5.

We propose PIVEN (**p**rediction **i**ntervals with specific **v**alue pr**e**dictio**n**), a novel approach for simultaneous PI generation and value prediction using DNNs. Our approach combines the benefits of both above-mentioned groups by producing *both* a PI and a value prediction, while ensuring that the latter is within the former. We follow the experimental procedure of recent works, and compare our approach to current best-performing methods: Quality-Driven PI (QD) by Pearce et al. (2018) (a dedicated PI generation method), and Deep Ensembles (DE) by DeepMind (Lakshminarayanan et al., 2017). Our results show that PIVEN outperforms QD by producing narrower PIs, while

simultaneously achieving comparable results to DE in terms of value prediction. Additional analysis on large image datasets and synthetic data shows that PIVEN performs well on skewed value distributions and can be effectively combined with pre-trained DNN architectures.

## 2 RELATED WORK

Modeling uncertainty in deep learning has been an active area of research in recent years (Pearce et al., 2018; Qiu et al., 2019; Gal & Ghahramani, 2016; Lakshminarayanan et al., 2017; Keren et al., 2018; Kendall & Gal, 2017; Geifman et al., 2018; Ovadia et al., 2019). Studies in uncertainty modeling and regression can be generally divided into two groups: *PI-based* and *non-PI-based*. Non-PI approaches utilize both Bayesian (MacKay, 1992) and non-Bayesian approaches. The former methods define a prior distribution on the weights and biases of a neural net (NN), while inferring a posterior distribution from the training data. Non-Bayesian methods (Gal & Ghahramani, 2016; Lakshminarayanan et al., 2017; Qiu et al., 2019) do not use initial prior distributions. In (Gal & Ghahramani, 2016), Monte Carlo sampling was used to estimate the predictive uncertainty of NNs through the use of dropout over multiple runs. A later study (Lakshminarayanan et al., 2017) employed a combination of ensemble learning and adversarial training to quantify model uncertainty. In an expansion of a previously-proposed approach (Nix & Weigend, 1994), each NN was optimized to learn the mean and variance of the data, assuming a Gaussian distribution. Recently, Qiu et al. (2019) proposed a post-hoc procedure using Gaussian processes to measure uncertainty.

PI-based approaches are designed to produce a PI for each sample. Keren et al. (2018) propose a post-processing approach that considers the regression problem as one of classification, and uses the output of the final softmax layer to produce PIs. Tagasovska & Lopez-Paz (2019) propose the use of a loss function designed to learn all conditional quantiles of a given target variable. LUBE (Khosravi et al., 2010) consists of a loss function optimized for the creation of PIs, but has the caveat of not being able to use stochastic gradient descent (SGD) for its optimization. A recent study (Pearce et al., 2018) inspired by LUBE, proposed a loss function that is both optimized for the generation of PIs and can be optimized using SGD.

Each of the two groups presented above tends to under-perform when applied to tasks for which its loss function was not optimized: Non-PI approaches produce more accurate value predictions, but are not optimized to produce PI and therefore produce bounds that are less tight. PI-based methods produce tight bounds, but tend to underperform when producing value predictions. . Recent studies (Kivaranovic et al., 2019; Romano et al., 2019) attempted to produce both value predictions and PIs by using conformal prediction with quantile regression. While effective, these methods use a complex splitting strategy, where one part of the data is used to produce value predictions and PIs, while the the other part is to further adjust the PIs. Recently, Salem et al. (2020) proposed a method for combining the two, together with post-hoc optimization. Contrary to these approaches, PIVEN produces PIs with value predictions in an end-to-end manner by relying on a novel loss function.

## 3 PROBLEM FORMULATION

In this work we consider a neural network regressor that processes an input $x \in \mathcal{X}$ with an associated label $y \in \mathbb{R}$, where $\mathcal{X}$ can be any feature space (e.g., tabular data, age prediction from images). Let $(x_i, y_i) \in \mathcal{X} \times \mathbb{R}$ be a data point along with its target value. Let $U_i$ and $L_i$ be the upper and lower bounds of PIs corresponding to the ith sample. Our goal is to construct $(L_i, U_i, y_i)$ so that $\Pr(L_i \leq y_i \leq U_i) \geq 1 - \alpha$. We refer to $1 - \alpha$ as the confidence level of the PI.

We define two quantitative measures for the evaluation of PIs, as defined in Khosravi et al. (2010). First, we define *coverage* as the ratio of dataset samples that fall within their respective PIs. We measure coverage using the *prediction interval coverage probability* (PICP) metric:

$$PICP := \frac{1}{n} \sum_{i=1}^{n} k_i \tag{1}$$

where $n$ denotes the number of samples and $k_i = 1$ if $y_i \in (L_i, U_i)$, otherwise $k_i = 0$. Next, we define a quality metric for the generated PIs with the goal of producing as tight a bound as possible

while maintaining adequate coverage. We define the *mean prediction interval width* (MPIW) as,

$$MPIW := \frac{1}{n} \sum_{i=1}^{n} U_i - L_i \tag{2}$$

When combined, these metrics enable us to comprehensively evaluate the quality of generated PIs.

## 4 METHOD

### 4.1 SYSTEM ARCHITECTURE

The proposed architecture is presented in Figure 1. It consists of three components:

**Backbone block.** The main body block, consisting of a varying number of DNN layers or sub-blocks. The goal of this component is to transform the input into a latent representation that is then provided as input to the other components. It is important to note that PIVEN supports any architecture type (e.g., dense, convolutions) that can be applied to a regression problem. Moreover, pre-trained architectures can also be used very easily, with PIVEN being added on top of the architecture for an additional short training. For example, we use pre-trained VGG-16 and DenseNet architectures in our experiments.

**Upper & lower-bound heads.** $L(x)$ and $U(x)$ produce the lower and upper bounds of the PI respectively, such that $\Pr(L(x) \leq y(x) \leq U(x)) \geq 1 - \alpha$ where $y(x)$ is the value prediction and $1 - \alpha$ is the predefined confidence level.

**Auxiliary head.** The auxiliary prediction head, $v(x)$, enables us to produce a value prediction. $v(x)$ does not produce the value prediction directly, but rather produces the *relative weight that should be given to each of the two bounds*. We define the value prediction using,

$$y = v \cdot U + (1 - v) \cdot L \tag{3}$$

where $v \in (0, 1)$. By expressing $y$ as a function of $U$ and $L$, we *(a)* bound them together and improve their performance (see Section 4.4), and; *(b)* ensure that the value prediction falls within the PI.

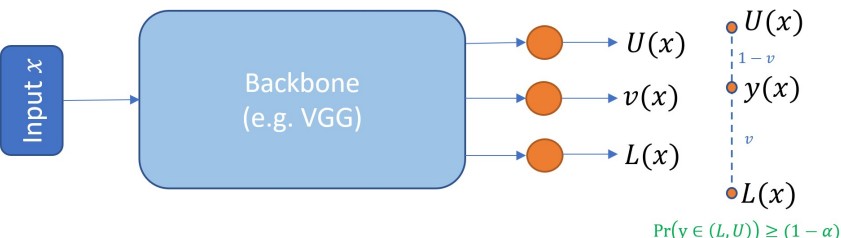

Figure 1: The PIVEN schematic architecture

The goal of the loss function of our approach, denoted as $\mathcal{L}_{PIVEN}$, is to balance two conflicting goals: generating tight PIs and producing accurate value predictions. $\mathcal{L}_{PIVEN}$ achieves this goal by synchronizing two loss functions that are designed to assess each of the two goals described above.

### 4.2 NETWORK OPTIMIZATION

Our goal is to generate narrow PIs, measured by *MPIW*, while maintaining the desired level of coverage $PICP = 1 - \alpha$. However, PIs that fail to capture their respective data point should not be encouraged to shrink further. We follow the derivation presented in Pearce et al. (2018) and define *captured MPIW* ($MPIW_{capt}$) as the $MPIW$ of only those points for which $L_i \leq y_i \leq U_i$,

$$MPIW_{capt} := \frac{1}{c} \sum_{i=1}^{n} (U_i - L_i) \cdot k_i \tag{4}$$

where $c = \sum_{i=1}^{n} k_i$. Hence, we seek to minimize $MPIW_{capt}$ subject to $PICP \geq 1 - \alpha$:

$$\theta^* = \arg\min_{\theta}(MPIW_{capt,\theta}) \ \ s.t \ \ PICP_{\theta} \geq 1 - \alpha$$

where $\theta$ is the parameters of the neural net. To enforce the coverage constraint, we utilize a variant of the well-known Interior Point Method (Potra & Wright, 2000), resulting in an unconstrained loss:

$$\mathcal{L}_{PI} = MPIW_{capt,\theta} + \sqrt{n} \cdot \lambda \Psi(1 - \alpha - PICP_{\theta})$$
$$\Psi(x) := \max(0, x)^2$$

where $\Psi$ is a quadratic penalty function, $n$ is the batch size which was included because a larger sample size would increase confidence in the value of PICP (thus increasing the loss) and $\lambda$ is a hyperparameter controlling the relative importance of width vs. coverage. We use a default value of $\lambda = 15$ in all our experiments, and perform further analyze of this parameter in Section 6.3

In practice, optimizing the loss with a discrete version of **k** (see eq. 4) fails to converge, because the gradient is always positive for all possible values. We therefore define a continuous version of **k**, denoted as $\mathbf{k}_{soft} = \sigma(s \cdot (\mathbf{y} - \mathbf{L})) \odot \sigma(s \cdot (\mathbf{U} - \mathbf{y}))$, where $\sigma$ is the sigmoid function, and $s > 0$ is a softening factor. The final version of $\mathcal{L}_{PI}$ uses the continuous and discrete versions of **k** in its calculations of the $PICP$ and $MPIW_{capt}$ metrics, respectively. The discrete **k** enables us to assign a score of zero to points outside the interval, while $\mathbf{k}_{soft}$ produces continuous values that enable gradient calculations.

Neural networks optimized by the abovementioned objective are able to generate well-calibrated PIs. However, they have two significant drawbacks—*overfitting*, and *lack of value prediction capabilities*.

**Overfitting.** The $MPIW_{capt}$ term in $\mathcal{L}_{PI}$ focuses only on the fraction $c$ of the training set where the data points are successfully captured by the PI. As a result, the network is likely to overfit to a subset of the data on which it was able to perform well in the first place. Our reasoning is supported by our experiments in sections 5 and 6.

**Lack of value prediction capabilities**. In its current form, $\mathcal{L}_{PI}$ is not able to perform value prediction, i.e., returning accurate prediction for the regression problem. To overcome this limitation, one can return the middle of the PI, as done in Pearce et al. (2018) and Tagasovska & Lopez-Paz (2019). This approach performs well only if the dataset value distribution is uniform or Gaussian (i.e., where choosing the mean minimizes prediction loss). For skewed distributions, as is often the case in the real world, this approach leads to sub-optimal results, as we show in Section 5.4.

We propose a novel loss function that combines the generation of both PIs and value predictions. To optimize the output of $v(x)$ (the auxiliary head), we minimize the standard regression loss,

$$\mathcal{L}_v = \frac{1}{n}\sum_{i=1}^{n} \ell(v_i \cdot U_i + (1 - v_i) \cdot L_i, \ y_i) \tag{5}$$

where $\ell$ is a regression objective against the ground-truth. Our final loss function is a convex combination of $\mathcal{L}_{PI}$, and the auxiliary loss $\mathcal{L}_v$. Thus, the overall training objective is:

$$\mathcal{L}_{PIVEN} = \beta\mathcal{L}_{PI} + (1 - \beta)\mathcal{L}_v \tag{6}$$

where $\beta$ is a hyperparameter that balances the two goals of our approach: producing narrow PIs and accurate value predictions. In all our experiments, we chose to assign equal priorities to both goals by setting $\beta = 0.5$. We performed no hyperparameter optimization.

### 4.3 USING ENSEMBLES TO ESTIMATE MODEL UNCERTAINTY

Each ensemble model produces a PI and a value prediction. These outputs need to be combined into a single PI and value prediction, thus capturing both the aleatoric and parametric uncertainties.

While PIs produced by a single PIVEN architecture could be used to capture aleatoric uncertainty, an ensemble of PIVEN architectures can capture the uncertainty of the PI itself. In this work, we consider the aggregation proposed by Lakshminarayanan et al. (2017), to follow fair evaluation as Pearce et al. (2018): Given an ensemble of $m$ NNs trained with $\mathcal{L}_{PIVEN}$, let $\tilde{U}, \tilde{L}$ represent the PI, and $\tilde{v}, \tilde{y}$ represent the ensemble's auxiliary and value prediction. We calculate

the PI uncertainty and use the ensemble to generate the PIs and value predictions as follows:

$$\bar{U}_i = \frac{1}{m} \sum_{j=1}^{m} U_{ij} \quad (7) \qquad \sigma_{PI}^2 = \sigma_{U_i}^2 = \frac{1}{m-1} \sum_{j=1}^{m} (U_{ij} - \bar{U}_i)^2 \quad (9)$$

$$\tilde{U}_i = \bar{U}_i + z_{\alpha/2} \cdot \sigma_{U_i} \quad (8) \qquad \tilde{y}_i = \frac{1}{m} \sum_{j=1}^{m} v_{ij} \cdot U_{ij} + (1 - v_{ij}) \cdot L_{ij} \quad (10)$$

where $(U_{ij} \ L_{ij})$ and $v_{ij}$ represents the PI and the auxiliary prediction for data point $i$, for NN $j$. A similar procedure is followed for $\tilde{L}_i$, subtracting $z_{\alpha/2} \cdot \sigma_{L_i}$, where $z_{\alpha/2}$ is the Z score for a confidence level $1 - \alpha$.

We found the above to perform satisfactorily in our experiments. However, PIVEN could potentially be further improved by incorporating aggregation methods such as Split Normal Mixture (SNM) (Wallis, 2014). We consider this venue as future work.

### 4.4 DISCUSSION OF CONTRIBUTIONS

PIVEN differs from previous studies in two important aspects. First, our approach is the first to propose an integrated architecture capable of producing both PIs and well-calibrated value predictions in an end-to-end manner, without using post-hoc steps. Moreover, since the auxiliary head produces predictions for all training set samples, it prevents PIVEN from overfitting to data points which were contained in their respective PIs, which is a possible problem for studies such as Pearce et al. (2018). This ability improves PIVEN's performance, as shown in Section 5.

PIVEN's second differentiating aspect is its method for producing the value prediction. While previous studies either provided the middle of the PI (Pearce et al., 2018; Tagasovska & Lopez-Paz, 2019) or the mean-variance (Lakshminarayanan et al., 2017) as their value predictions, PIVEN's auxiliary head can produce any value within the PI, while also being distribution free (i.e., without making any assumptions regarding data distribution within the PI). By expressing the value prediction as a function of the upper and lower bounds, we ensure that the three heads are synchronized and that the prediction *always* falls within the PI. Our use of a *non-Gaussian observation model* enables us to produce predictions that are not in the middle of the interval and create representations that are more characteristic of many real-world cases, where the points within the PI is not necessarily uniformly distributed. Our experiments, presented in Sections 5.4 and 5.5 support our conclusions.

## 5 EVALUATION

### 5.1 DATASETS

**UCI Datasets.** We conduct our experiments on a set of benchmark datasets used by several state-of-the-art studies (Gal & Ghahramani, 2016; Hernández-Lobato & Adams, 2015; Lakshminarayanan et al., 2017; Pearce et al., 2018). This benchmark includes ten datasets (Asuncion & Newman, 2007).

**IMDB age estimation dataset.** The IMDB-WIKI dataset (Rothe et al., 2018) is currently the largest age-labeled facial dataset available. Our dataset consists of 460,723 images from 20,284 celebrities, and the regression goal is to predict the age of the person in the image. It is important to note that this dataset is known to contain noise (i.e., aleatoric uncertainty), thus making it highly relevant to this study (despite usually being used for pre-training).

**RSNA pediatric bone age dataset.** This dataset is a popular medical imaging dataset consisting of X-ray images of children's hands (Halabi et al., 2019). The regression task is predicting one's age from one's bone image. The dataset contains 12,611 training images and 200 test set images.

While the first group of datasets enables us to compare PIVEN's performance to recent state-of-the-art studies in the field, the two latter datasets enable us to demonstrate that our approach is both scalable and effective on multiple types of input.

## 5.2 Baselines

We compare our performance to two top-performing NN-based baselines from recent years:

**Quality driven PI (QD) (Pearce et al., 2018).** This approach produces PIs that minimize a smooth combination of the PICP/MPIW metrics without considering the value prediction task in its objective function. Its reported results make this approach state-of-the-art in terms of PI width and coverage.

**Deep Ensembles (DE) (Lakshminarayanan et al., 2017).** This work combines individual conditional Gaussian distribution with adversarial training, and uses the models' variance to compute prediction intervals. Because DE outputs distributions instead of PIs, we first convert it to PIs, and then compute PICP and MPIW. Its reported results make this method one of the top performers with respect to the RMSE metric (i.e., value prediction).

By comparing PIVEN to these two baselines, we are able to evaluate its ability to simultaneously satisfy the two main requirements for regression problems in domains with high certainty.

## 5.3 Experimental Setup

Throughout our experiments, we evaluate our two baselines using their reported deep architectures and hyperparmeters. For full experimental details, please see Appendix A. We ran our experiments using a GPU server with two NVIDIA Tesla P100. Our code is implemented using TensorFlow and Keras (Abadi et al., 2016; Chollet et al., 2015), and made available online[1].

**UCI datasets.** We use the experimental setup proposed by Hernández-Lobato & Adams (2015), which was also used by our baselines. Results are averaged on 20 random 90%/10% splits of the data, except for the "Year Prediction MSD" and "Protein", which were split once and five times respectively. Our network architecture is identical to that of our two baselines: one hidden layer with ReLU activation function (Nair & Hinton, 2010), the Adam optimizer (Kingma & Ba, 2014), and ensemble size $M = 5$. Input and target variables are normalized to zero mean and unit variance.

**IMDB age estimation dataset.** We use the DenseNet architecture (Huang et al., 2017) as the backbone block, then add two dense layers. We apply the data preprocessing used in (Yang et al., 2018). We report the results for 5-fold cross validation.

**RSNA bone age dataset.** We use the VGG-16 architecture (Simonyan & Zisserman, 2014) as the backbone block, with weights pre-trained on ImageNet. We add two convolutional layers followed by a dense layer, and perform additional training. This dataset has 200 predefined test images.

## 5.4 UCI Datasets Evaluation Results and Analysis

We use two evaluation metrics: MPIW and RMSE, with results presented in Table 1. The desired coverage, measured by PICP, was set to 95%. In terms of PI-quality, PIVEN outperforms QD in nine out of ten datasets (although no method reached the required PICP in the "Boston" and "Concrete" datasets, and therefore not in bold for DE), while achieving equal performance in the remaining dataset. DE trails behind PIVEN and QD in most datasets, which is to be expected since this approach does not attempt to optimize MPIW. For the RMSE metric, it is clear that PIVEN and DE are the top performers, with the former achieving the best results in five datasets, and the latter in four. The QD baseline trails behind the other methods in all datasets but one. QD's performance is not surprising given that the focus of this approach is PI generation rather than value prediction. We also perform additional experiments for DE with ensemble size $M = 1$. See full results in Table 6 in the appendix.

The results of our experiments clearly show that PIVEN is capable of providing accurate value predictions for regression problems (i.e., achieving competitive results with the top-performing DE baseline) while achieving state-of-the-art results in uncertainty modeling by the use of PIs. Please note that higher $\alpha$ values (i.e., lower PICP) enable our model to significantly outperform the baselines in other metrics. Please see our analysis in Section 6.2

**Analysis.** In Section 4.4 we describe our rationale in expressing the value prediction as a function of the upper and lower bounds of the interval. To further explore the benefits of our auxiliary head $v$, we evaluate two variants of PIVEN. In the first variant, denoted as POO (point-only optimization),

---

[1] https://github.com/anonymous/anonymous

we decouple the value prediction from the PI. The loss function of this variant is $\mathcal{L}_{PI} + \ell(v, y_{true})$ where $\ell$ is set to be MSE loss. In the second variant, denoted MOI (middle of interval), the value prediction produced by the model is always the middle of the PI (in other words, $v$ is set to 0.5). While both return the middle of the PI as point prediction, MOI differs from QD by an additional component in its loss function.

The results of our analysis are presented in Table 2, which contains the results of the MPIW and RMSE metrics (the PICP values are identical for all variants and are therefore omitted—see full results in Appendix B). The full PIVEN significantly outperforms the two other variants. This leads us to conclude that both novel aspects of our approach—the simultaneous optimization of PI-width and RMSE, and the ability to select any value on the PI as the value prediction—contribute to PIVEN's performance. Finally, it is important to note that while inferior to PIVEN, both the POO and MOI variants outperform the QD baseline in terms of MPIW, while being equal or better for RMSE. These results, and in particular the comparison of POO to PIVEN with respect to their RMSE values, provide a clear motivation for deploying the full version of PIVEN.

Table 1: Regression results for benchmark UCI datasets comparing PICP, MPIW, and RMSE. Top performance defined as in Pearce et al. (2018): every approach with PICP $\geq 0.95$ was defined as best for PICP. For MPIW, lowest value was best. If PICP $\geq 0.95$ for neither, the largest PICP was best, and MPIW was only assessed if the one with larger PICP also had smallest MPIW.

| | PICP | | | MPIW | | | RMSE | | |
|---|---|---|---|---|---|---|---|---|---|
| Datasets | DE | QD | PIVEN | DE | QD | PIVEN | DE | QD | PIVEN |
| Boston | $0.87 \pm 0.01$ | $\mathbf{0.93 \pm 0.01}$ | $\mathbf{0.93 \pm 0.01}$ | $0.87 \pm 0.03$ | $1.15 \pm 0.02$ | $1.09 \pm 0.01$ | $\mathbf{2.87 \pm 0.19}$ | $3.39 \pm 0.26$ | $3.13 \pm 0.21$ |
| Concrete | $0.92 \pm 0.01$ | $\mathbf{0.93 \pm 0.01}$ | $\mathbf{0.93 \pm 0.01}$ | $1.01 \pm 0.02$ | $1.08 \pm 0.01$ | $1.02 \pm 0.01$ | $\mathbf{5.21 \pm 0.09}$ | $5.88 \pm 0.10$ | $5.43 \pm 0.13$ |
| Energy | $\mathbf{0.99 \pm 0.00}$ | $0.97 \pm 0.01$ | $0.97 \pm 0.00$ | $0.49 \pm 0.01$ | $0.45 \pm 0.01$ | $\mathbf{0.42 \pm 0.01}$ | $1.68 \pm 0.06$ | $2.28 \pm 0.04$ | $\mathbf{1.65 \pm 0.03}$ |
| Kin8nm | $\mathbf{0.97 \pm 0.00}$ | $0.96 \pm 0.00$ | $0.96 \pm 0.00$ | $1.14 \pm 0.01$ | $1.18 \pm 0.00$ | $\mathbf{1.10 \pm 0.00}$ | $0.08 \pm 0.00$ | $0.08 \pm 0.00$ | $\mathbf{0.07 \pm 0.00}$ |
| Naval | $\mathbf{0.98 \pm 0.00}$ | $0.97 \pm 0.00$ | $\mathbf{0.98 \pm 0.00}$ | $0.31 \pm 0.01$ | $0.27 \pm 0.00$ | $\mathbf{0.24 \pm 0.00}$ | $0.00 \pm 0.00$ | $\mathbf{0.00 \pm 0.00}$ | $0.00 \pm 0.00$ |
| Power | $\mathbf{0.96 \pm 0.00}$ | $0.96 \pm 0.00$ | $0.96 \pm 0.00$ | $0.91 \pm 0.00$ | $\mathbf{0.86 \pm 0.00}$ | $\mathbf{0.86 \pm 0.00}$ | $3.99 \pm 0.04$ | $4.14 \pm 0.04$ | $4.08 \pm 0.04$ |
| Protein | $\mathbf{0.96 \pm 0.00}$ | $0.95 \pm 0.00$ | $0.95 \pm 0.00$ | $2.68 \pm 0.01$ | $\mathbf{2.27 \pm 0.01}$ | $2.26 \pm 0.01$ | $\mathbf{4.36 \pm 0.02}$ | $4.99 \pm 0.02$ | $4.35 \pm 0.02$ |
| Wine | $0.90 \pm 0.01$ | $\mathbf{0.91 \pm 0.01}$ | $\mathbf{0.91 \pm 0.01}$ | $2.50 \pm 0.02$ | $2.24 \pm 0.02$ | $\mathbf{2.22 \pm 0.01}$ | $\mathbf{0.62 \pm 0.01}$ | $0.67 \pm 0.01$ | $0.63 \pm 0.01$ |
| Yacht | $\mathbf{0.98 \pm 0.01}$ | $0.95 \pm 0.01$ | $0.95 \pm 0.01$ | $0.33 \pm 0.02$ | $0.18 \pm 0.00$ | $\mathbf{0.17 \pm 0.00}$ | $1.38 \pm 0.07$ | $1.10 \pm 0.06$ | $\mathbf{0.98 \pm 0.07}$ |
| MSD | $\mathbf{0.95 \pm NA}$ | $\mathbf{0.95 \pm NA}$ | $\mathbf{0.95 \pm NA}$ | $2.91 \pm NA$ | $2.45 \pm NA$ | $\mathbf{2.42 \pm NA}$ | $8.95 \pm NA$ | $9.30 \pm NA$ | $\mathbf{8.93 \pm NA}$ |

Table 2: Analysis comparing MPIW and RMSE. Results were analyzed as in Table 1.

| | MPIW | | | RMSE | | |
|---|---|---|---|---|---|---|
| Datasets | POO | MOI | PIVEN | POO | MOI | PIVEN |
| Boston | $\mathbf{1.09 \pm 0.02}$ | $1.15 \pm 0.02$ | $\mathbf{1.09 \pm 0.01}$ | $3.21 \pm 0.24$ | $3.39 \pm 0.27$ | $\mathbf{3.13 \pm 0.21}$ |
| Concrete | $\mathbf{1.02 \pm 0.01}$ | $1.07 \pm 0.01$ | $\mathbf{1.02 \pm 0.01}$ | $5.55 \pm 0.11$ | $5.73 \pm 0.10$ | $\mathbf{5.43 \pm 0.13}$ |
| Energy | $\mathbf{0.42 \pm 0.01}$ | $0.45 \pm 0.01$ | $\mathbf{0.42 \pm 0.01}$ | $2.16 \pm 0.04$ | $2.27 \pm 0.04$ | $\mathbf{1.65 \pm 0.03}$ |
| Kin8nm | $1.13 \pm 0.00$ | $1.17 \pm 0.00$ | $\mathbf{1.10 \pm 0.00}$ | $0.08 \pm 0.00$ | $0.08 \pm 0.00$ | $\mathbf{0.07 \pm 0.00}$ |
| Naval | $\mathbf{0.24 \pm 0.00}$ | $0.30 \pm 0.02$ | $\mathbf{0.24 \pm 0.00}$ | $0.00 \pm 0.00$ | $0.00 \pm 0.00$ | $0.00 \pm 0.00$ |
| Power | $\mathbf{0.86 \pm 0.00}$ | $\mathbf{0.86 \pm 0.00}$ | $\mathbf{0.86 \pm 0.00}$ | $4.13 \pm 0.04$ | $4.15 \pm 0.04$ | $\mathbf{4.08 \pm 0.04}$ |
| Protein | $\mathbf{2.25 \pm 0.01}$ | $2.27 \pm 0.01$ | $2.26 \pm 0.01$ | $4.78 \pm 0.02$ | $4.99 \pm 0.02$ | $\mathbf{4.35 \pm 0.02}$ |
| Wine | $2.24 \pm 0.01$ | $2.23 \pm 0.01$ | $\mathbf{2.22 \pm 0.01}$ | $0.64 \pm 0.01$ | $0.67 \pm 0.01$ | $\mathbf{0.63 \pm 0.01}$ |
| Yacht | $0.18 \pm 0.00$ | $0.19 \pm 0.01$ | $\mathbf{0.17 \pm 0.00}$ | $0.99 \pm 0.07$ | $1.15 \pm 0.08$ | $\mathbf{0.98 \pm 0.07}$ |
| MSD | $\mathbf{2.42 \pm NA}$ | $2.43 \pm NA$ | $\mathbf{2.42 \pm NA}$ | $9.10 \pm NA$ | $9.25 \pm NA$ | $\mathbf{8.93 \pm NA}$ |

## 5.5 LARGE-SCALE IMAGE DATASETS

We now evaluate PIVEN on two large and noisy image datasets: bone age and age estimation. We add an additional baseline – denoted as NN – of a dense layer on top of an existing architecture (DenseNet/VGG, see Section 5.3), that outputs value prediction using the MSE metric. In doing so, we follow the approach used in Qiu et al. (2019) for similar evaluation.

Our results are presented in Table 3. We use *mean absolute error* (MAE), which was the datasets' chosen metric. For the IMDB age prediction dataset, results show that PIVEN outperforms both baselines across all metrics. It is particularly noteworthy that our approach achieves *both* higher coverage and tighter PIs compared to QD. We attribute the significant improvement in MPIW – 17% – to the fact that this dataset has relatively high degrees of noise (Yang et al., 2018). In the bone age dataset, PIVEN outperforms both baselines in terms of MAE. Our approach fares slightly worse compared to QD on the MPIW metric, but that is likely due to its higher coverage (i.e., PICP).

Our results support our hypothesis that for large and high-dimensional data (and in particular those with high degrees of noise), PIVEN is likely to outperform previous work due to its ability to combine

value predictions with PI generation. PIVEN produces tighter PIs and place the value prediction more accurately within the PI. A detailed analysis of the training process – in terms of training/validation loss, MAE, PICP and MPIW – is presented in Appendix E and further supports our conclusions. Please note that a major reason for DE's high performance is the fact that population age distribution is a Gaussian, and is therefore fully compatible with this baseline's assumptions. PIVEN is able to obtain comparable results to DE without relying on any distributional assumptions.

Table 3: Results on the RSNA bone age and IMDB age estimation datasets.

| Datasets | Method | PICP | MPIW | MAE |
|---|---|---|---|---|
| Bone age | NN | NA | NA | 18.68 |
| | PIVEN | **0.93** | 2.09 | **18.13** |
| | QD | 0.9 | **1.99** | 20.24 |
| | DE | **0.93** | 2.17 | 18.69 |
| IMDB age | NN | NA | NA | $7.08 \pm 0.03$ |
| | PIVEN | **0.95 $\pm$ 0.01** | $2.87 \pm 0.04$ | $7.03 \pm 0.04$ |
| | QD | $0.92 \pm 0.01$ | $3.47 \pm 0.03$ | $10.23 \pm 0.12$ |
| | DE | **0.95 $\pm$ 0.01** | **2.61 $\pm$ 0.05** | **6.66 $\pm$ 0.06** |

Additionally, we evaluated PIVEN's performance under dataset shift, using the Flight Delay dataset (Hensman et al., 2013). Our results are presented in Appendix C.

# 6 DISCUSSION

## 6.1 VALUE PREDICTION IN NON-UNIFORM PI DISTRIBUTIONS

As noted in Pearce et al. (2018), the QD baseline's approach of producing value predictions inside the PI "breaks the distribution-free assumption". This means that for value distributions that are neither uniform nor Gaussian, QD's approach of returning the middle of the interval as the value is likely sub-optimal. To test this hypothesis, we evaluate PIVEN and QD on the Sine and skewed-normal distributions. The results, presented in Figure 2 (for skewed-normal) and Appendix A.1 (for Sine), clearly illustrate PIVEN's superior ability to adapt to multiple value distributions.

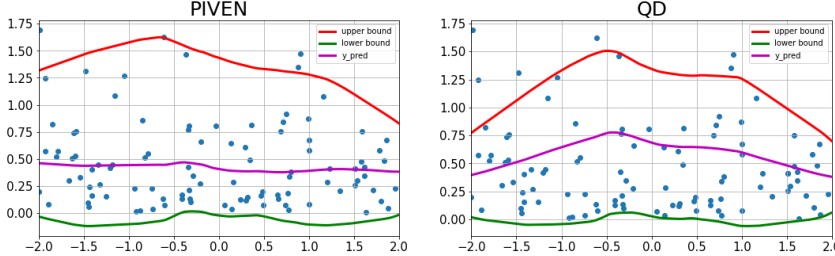

Figure 2: Value prediction comparison, where $y$ is sampled from skewed-normal distribution $f(x; \alpha) = 2\phi(x)\Phi(\alpha \cdot x)$ where $\phi(x)$ and $\Phi(\alpha \cdot x)$ denote the $\mathcal{N}(0, 1)$ density and distribution function respectively; the parameter $\alpha$ denotes the skewness parameter.

## 6.2 PI PERFORMANCE AS A FUNCTION OF COVERAGE

In Section 4.4 we argue that PIVEN's auxiliary head has the secondary effect of enabling it to train on the entire training set rather than only on the data points captured within their PIs. We argue that this property makes PIVEN more robust compared to current SOTA benchmarks. We explore the effects of capturing a smaller percentage of data points (i.e., smaller coverage) on the performance of both PIVEN and QD. Given that QD ignores such data points, it will optimize its performance for a smaller training set. A summary of the MPIW results is presented in Table 4 and full results are in Appendix D. As the coverage decreases, PIVEN's relative performance to QD continually improves. Interestingly, PIVEN's RMSE performance remains unchanged (Figure 3). The results clearly support our hypothesis regarding PIVEN's training robustness to points outside the PI.

| Datasets | alpha | | | | | |
| | 0.05 | 0.10 | 0.15 | 0.20 | 0.25 | 0.30 |
|---|---|---|---|---|---|---|
| Boston | 7% | 7% | 6% | 7% | 8% | 8% |
| Concrete | 7% | 7% | 8% | 7% | 12% | 13% |
| Energy | 6% | 12% | 8% | 17% | 19% | 27% |
| Kin8nm | 5% | 7% | 9% | 19% | 13% | 15% |
| Naval | 10% | 16% | 16% | 14% | 17% | 23% |
| Power | 0% | 0% | 1% | 1% | 1% | 1% |
| Protein | 1% | 1% | 2% | 3% | 2% | 2% |
| Wine | 0% | 1% | 2% | 2% | 4% | 4% |
| Yacht | 8% | 12% | 14% | 20% | 20% | 21% |
| MSD | 1% | -3% | -3% | 1% | -5% | -2% |
| **Mean** | **4.5%** | **6.0%** | **6.3%** | **9.1%** | **9.1%** | **11.2%** |

Table 4: MPIW improvement in percentages (PIVEN relative to QD) as the coverage decreases (i.e, alpha increases).

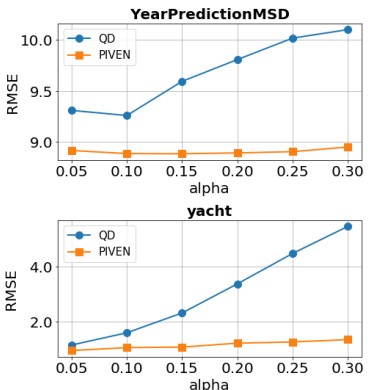

Figure 3: RMSE as function of $\alpha$ on two of the UCI datasets. See Appendix D for full results.

## 6.3 HYPERPARAMETERS INFLUENCE

We now analyze the effects of selecting various values to the two parameters that govern the behavior of our approach: $\beta$ and $\lambda$. We analyze the behavior of our approach using the Sine function, whose value fluctuations are challenging to any PI-producing approach.

The goal of $\beta$ is to balance the two goals of our approach: producing narrow PIs and accurate value predictions. For large values of this parameter, we expect PIVEN to put greater emphasis on optimizing the PI at the expense of the value prediction. For small values of $\beta$, we expect the opposite. Figure 4 (Left) confirms our expectations. It is clear that $\beta = 0.99$ produces the most accurate PIs, but its value predictions is the least accurate. The opposite applies for $\beta = 0.1$, which produces the most accurate value predictions but whose PIs are the least accurate. This analysis also shows that $\beta = 0.5$ strikes a better balance than the other settings, as it optimizes for both tasks simultaneously.

The goal of $\lambda$ is to balance the goal of capturing as many samples as possible within our PIs (PICP) with the conflicting goal of producing as tight a PI as possible for each sample (MPIW). We expect that for small values of $\lambda$, PIVEN will attempt to produce tight PIs with less emphasis on coverage. For large $\lambda$ values, we expect the opposite. Results in Figure 4 confirm our expectations.

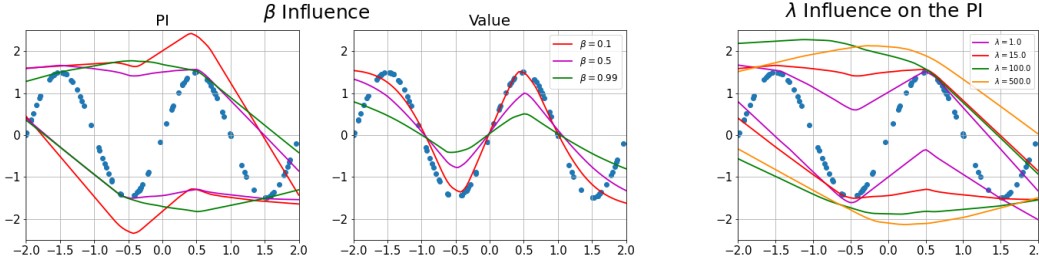

Figure 4: (Left) An analysis of the effect of varying $\beta$ values on PIVEN's performance in terms of PI width and value prediction. (Right) The effect of various $\lambda$ values on the tightness of the PI

## 7 CONCLUSIONS

We present PIVEN, a novel architecture for combining the generation of prediction intervals together with specific value predictions in an end-to-end manner. By optimizing for these two goals simultaneously we are able to produce tighter intervals while at the same time achieving greater quality in our value predictions. PIVEN's performance is enhanced by two factors. First, its value-prediction and PI-prediction are connected, making it better at modeling non-uniform distributions in the PI. Secondly, PIVEN does not ignore but rather trains on data points that are not captured by the PI.

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

## A    EXPERIMENTAL SETUP

In this section we provide full details of our dataset preprocessing and experiments presented in the main study. Our code is available online [2]

### A.1    SYNTHETIC DATASETS

For the qualitative training method comparison, QD vs. PIVEN (Section 6.1), all NNs used ReLU activations and 100 nodes in one hidden layer. Both methods trained until convergence using the same initialization, loss and default hyperarameters, as described in Pearce et al. (2018): $\alpha = 0.05, \beta = 0.5, \lambda = 15.0, s = 160.0$, and the random seed was set to 1. The generated data consisted of 100 points sampled uniformly from the interval $[-2, 2]$. We used $\alpha = 100$ as the skewness parameter. In Figures 5 and 6 we present a comparison on the Sine and skewed-normal distributions.

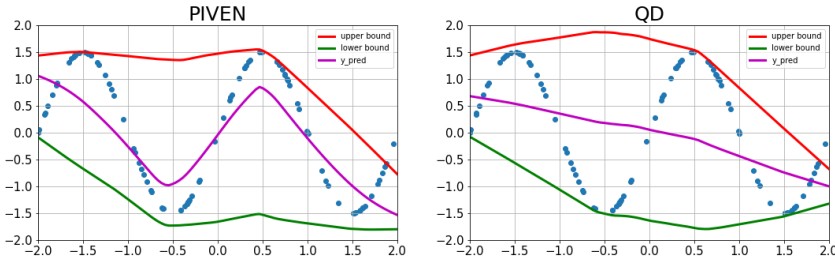

Figure 5: Comparison of value prediction for QD vs. PIVEN method, where $\boldsymbol{y}$ was generated by $f(x) = 1.5 \sin(x)$.

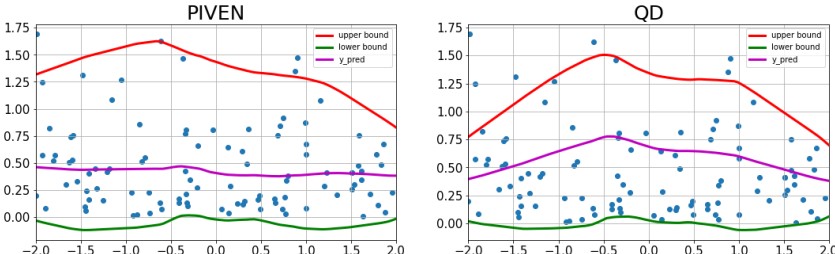

Figure 6: Value prediction comparison, where $\boldsymbol{y}$ is sampled from skewed-normal distribution $f(x; \alpha) = 2\phi(x)\Phi(\alpha \cdot x)$ where $\phi(x)$ and $\Phi(\alpha \cdot x)$ denote the $\mathcal{N}(0, 1)$ density and distribution function respectively; the parameter $\alpha$ denotes the skewness parameter.

### A.2    DATASET PREPROCESSING AND FULL EXPERIMENTAL SETUP

In addition to the ten benchmark datasets used by all recent studies in the field, we evaluated PIVEN on two large image datasets. Due to the size of the datasets and the nature of the domain, preprocessing was required. We provide the full details of the process below.

**UCI datasets.** For the UCI datasets, we used the experimental setup proposed by Hernández-Lobato & Adams (2015), which was also used in all the two baselines described in this study. All datasets were averaged on 20 random splits of the data, except for the "Year Prediction MSD" and "protein" datasets. Since "Year Prediction MSD" has predefined fixed splits by the provider, only one run was conducted. For "protein", 5 splits were used, as was done in previous work. We used identical network architectures to those described in previous works: one dense layer with ReLU (Nair & Hinton, 2010), containing 50 neurons for each network. In the "Year Prediction MSD" and "protein" datasets where NNs had 100 neurons. Regarding train/test split and hyperparameters, we employ the same setup as Pearce et al. (2018): train/test folds were randomly split 90%/10%, input and

---

[2]https://github.com/anonymous/anonymous

target variables were normalized to zero mean and unit variance. The softening factor was constant for all datasets, $s = 160.0$. For the majority of the datasets we used $\lambda = 15.0$, except for "naval", "protein", "wine" and "yacht" where $\lambda$ was set to 4.0, 40.0, 30.0 and 3.0 respectively. The value of the parameter $\beta$ was set to 0.5. The Adam optimizer (Kingma & Ba, 2014) was used with exponential decay, where learning rate and decay rate were tuned. Batch size of 100 was used for all the datasets, except for "Year Prediction MSD" where batch size was set to 1000. Five neural nets were used in each ensemble, using parameter re-sampling. The objective used to optimized $v$ was Mean Square Error (MSE) for all datasets. We also tune $\lambda$, initializing variance, and number of training epochs using early stopping. To ensure that our comparison with the state-of-the-art baselines is accurate, we first set the parameters of our neural nets so that they produce the results reported in Pearce et al. (2018). We then use the same parameter configurations in our experiments of PIVEN.

**IMDB age estimation dataset**[3]. For the IMDB dataset, we used the DenseNet architecture (Huang et al., 2017) as a feature extractor. On top of this architecture we added two dense layers with dropout. The sizes of the two dense layers were 128 and 32 neurons respectively, with a dropout factor of 0.2, and ReLU activation (Nair & Hinton, 2010). In the last layer, the biases of the PIs were initially set to $[5.0, -5.0]$ for the upper and lower bounds respectively. We used the data preprocessing similar to that of previous work (Yang et al., 2018; Zhang et al., 2017): all face images were aligned using facial landmarks such as eyes and the nose. After alignment, the face region of each image was cropped and resized to a $64 \times 64$ resolution. In addition, common data augmentation methods, including zooming, shifting, shearing, and flipping were randomly activated. The Adam optimization method (Kingma & Ba, 2014) was used for optimizing the network parameters over 90 epochs, with a batch size of 128. The learning rate was set to 0.002 initially and reduced by a factor 0.1 every 30 epochs. Regarding loss hyperparameters, we used the standard configuration proposed in Pearce et al. (2018): confidence interval set to 0.95, soften factor set to 160.0 and $\lambda = 15.0$. For PIVEN we used the same setting, with $\beta = 0.1$. Since there was no predefined test set for this dataset, we employed a 5-fold cross validation: In each split, we used 20% as the test set. Additionally, 20% of the train set was designated as the validation set. Best model obtained by minimizing the validation loss. In QD and PIVEN, we normalized ages to zero mean and unit variance.

**RSNA pediatric bone age dataset**[4]. For the RSNA dataset, we used the well-known VGG-16 architecture (Simonyan & Zisserman, 2014) as a base model, with weights pre-trained on ImageNet. On top of this architecture, we added batch normalization (Ioffe & Szegedy, 2015), attention mechanism with two CNN layers of 64 and 16 neurons each, two average pooling layers, dropout (Srivastava et al., 2014) with a 0.25 probability, and a fully connected layer with 1024 neurons. The activation function for the CNN layers was ReLU (Nair & Hinton, 2010), and we used ELU for the fully connected layer. For the PIs last layer, we used biases of $[2.0, -2.0]$, for the upper and lower bound initialization, respectively. We used standard data augmentation consisting of horizontal flips, vertical and horizontal shifts, and rotations. In addition, we normalized targets to zero mean and unit variance. To reduce computational costs, we downscaled input images to $384 \times 384$ pixels. The network was optimized using Adam optimizer (Kingma & Ba, 2014), with an initial learning rate of 0.01 which was reduced when the validation loss has stopped improving over 10 epochs. We trained the network for 50 epochs using batch size of 100. For our loss hyperparameters, we used the standard configuration like proposed in Pearce et al. (2018): confidence interval set to 0.95, soften factor set to 160.0 and $\lambda = 15.0$. For PIVEN, we used the same setting, with $\beta = 0.5$.

---

[3]https://data.vision.ee.ethz.ch/cvl/rrothe/imdb-wiki/
[4]https://www.kaggle.com/kmader/rsna-bone-age

## B ANALYSIS FULL RESULTS

We now present the full results of our ablation studies, including PICP, for the ablation variants:

Table 5: Ablation analysis, comparing PICP, MPIW and RMSE. The guidelines for selecting best results are as those used in Table 1.

| Datasets | PICP | | | MPIW | | | RMSE | | |
|---|---|---|---|---|---|---|---|---|---|
| | POO | MOI | PIVEN | POO | MOI | PIVEN | POO | MOI | PIVEN |
| Boston | **0.93 ± 0.01** | **0.93 ± 0.01** | **0.93 ± 0.01** | **1.09 ± 0.02** | 1.15 ± 0.02 | **1.09 ± 0.01** | 3.21 ± 0.24 | 3.39 ± 0.27 | **3.13 ± 0.21** |
| Concrete | **0.93 ± 0.01** | **0.93 ± 0.01** | **0.93 ± 0.01** | **1.02 ± 0.01** | 1.07 ± 0.01 | **1.02 ± 0.01** | 5.55 ± 0.11 | 5.73 ± 0.10 | **5.43 ± 0.13** |
| Energy | **0.97 ± 0.01** | **0.97 ± 0.00** | **0.97 ± 0.00** | **0.42 ± 0.01** | 0.45 ± 0.01 | **0.42 ± 0.01** | 2.16 ± 0.04 | 2.27 ± 0.04 | **1.65 ± 0.03** |
| Kin8nm | **0.96 ± 0.00** | **0.96 ± 0.00** | **0.96 ± 0.00** | 1.13 ± 0.00 | 1.17 ± 0.00 | **1.10 ± 0.00** | 0.08 ± 0.00 | 0.08 ± 0.00 | **0.07 ± 0.00** |
| Naval | **0.98 ± 0.00** | **0.98 ± 0.00** | **0.98 ± 0.00** | **0.24 ± 0.00** | 0.30 ± 0.02 | **0.24 ± 0.00** | **0.00 ± 0.00** | **0.00 ± 0.00** | **0.00 ± 0.00** |
| Power | **0.96 ± 0.00** | **0.96 ± 0.00** | **0.96 ± 0.00** | **0.86 ± 0.00** | **0.86 ± 0.00** | **0.86 ± 0.00** | 4.13 ± 0.04 | 4.15 ± 0.04 | **4.08 ± 0.04** |
| Protein | **0.95 ± 0.00** | **0.95 ± 0.00** | **0.95 ± 0.00** | **2.25 ± 0.01** | 2.27 ± 0.01 | 2.26 ± 0.01 | 4.78 ± 0.02 | 4.99 ± 0.01 | **4.35 ± 0.02** |
| Wine | **0.91 ± 0.01** | **0.91 ± 0.01** | **0.91 ± 0.01** | 2.24 ± 0.01 | 2.23 ± 0.01 | **2.22 ± 0.01** | 0.64 ± 0.01 | 0.67 ± 0.01 | **0.63 ± 0.01** |
| Yacht | **0.95 ± 0.01** | **0.95 ± 0.01** | **0.95 ± 0.01** | 0.18 ± 0.01 | 0.19 ± 0.01 | **0.17 ± 0.00** | 0.99 ± 0.07 | 1.15 ± 0.08 | **0.98 ± 0.07** |
| MSD | **0.95 ± NA** | **0.95 ± NA** | **0.95 ± NA** | **2.42 ± NA** | 2.43 ± NA | **2.42 ± NA** | 9.10 ± NA | 9.25 ± NA | **8.93 ± NA** |

Additionally, we tested the Deep Ensembles baseline with an ensemble size of one. We expect the baseline to under-perform in this setting, and this expectation is supported by the results presented in Table 6.

Table 6: Results on regression benchmark UCI datasets comparing PICP, MPIW, and RMSE. The guidelines for selecting best results are as those used in Table 1. Note: the results for DE are presented for ensemble size of one, denotes as *DE-One*.

| Datasets | PICP | | | MPIW | | | RMSE | | |
|---|---|---|---|---|---|---|---|---|---|
| | DE-One | QD | PIVEN | DE-One | QD | PIVEN | DE-One | QD | PIVEN |
| Boston | 0.76 ± 0.01 | **0.93 ± 0.01** | **0.93 ± 0.01** | 0.75 ± 0.02 | 1.15 ± 0.02 | 1.09 ± 0.01 | **3.11 ± 0.21** | 3.39 ± 0.26 | 3.13 ± 0.21 |
| Concrete | 0.87 ± 0.01 | **0.93 ± 0.01** | **0.93 ± 0.01** | 0.93 ± 0.02 | 1.08 ± 0.01 | 1.02 ± 0.01 | 5.52 ± 0.13 | 5.88 ± 0.10 | **5.43 ± 0.13** |
| Energy | 0.93 ± 0.01 | **0.97 ± 0.01** | **0.97 ± 0.00** | 0.43 ± 0.02 | 0.45 ± 0.01 | **0.42 ± 0.01** | 1.72 ± 0.06 | 2.28 ± 0.04 | **1.65 ± 0.05** |
| Kin8nm | 0.93 ± 0.00 | **0.96 ± 0.00** | **0.96 ± 0.00** | 1.06 ± 0.01 | 1.18 ± 0.00 | **1.10 ± 0.00** | 0.08 ± 0.00 | 0.08 ± 0.00 | **0.07 ± 0.00** |
| Naval | 0.94 ± 0.00 | **0.97 ± 0.00** | **0.98 ± 0.00** | 0.25 ± 0.01 | 0.27 ± 0.00 | **0.24 ± 0.00** | **0.00 ± 0.00** | **0.00 ± 0.00** | **0.00 ± 0.00** |
| Power | **0.96 ± 0. 00** | **0.96 ± 0.00** | **0.96 ± 0.00** | 0.90 ± 0.00 | **0.86 ± 0.00** | **0.86 ± 0.00** | 4.01 ± 0.04 | 4.14 ± 0.04 | 4.08 ± 0.04 |
| Protein | **0.95 ± 0.00** | **0.95 ± 0.00** | **0.95 ± 0.00** | 2.64 ± 0.01 | 2.27 ± 0.01 | 2.26 ± 0.01 | 4.43 ± 0.02 | 4.99 ± 0.02 | **4.35 ± 0.02** |
| Wine | 0.86 ± 0.01 | **0.91 ± 0.01** | **0.91 ± 0.01** | 2.34 ± 0.02 | 2.24 ± 0.02 | **2.22 ± 0.01** | 0.64 ± 0.01 | 0.67 ± 0.01 | **0.63 ± 0.01** |
| Yacht | **0.95 ± 0.01** | **0.95 ± 0.01** | **0.95 ± 0.01** | 0.26 ± 0.02 | 0.18 ± 0.00 | **0.17 ± 0.00** | 1.42 ± 0.07 | 1.10 ± 0.06 | **0.98 ± 0.07** |
| MSD | **0.95 ± NA** | **0.95 ± NA** | **0.95 ± NA** | 2.86 ± NA | 2.45 ± NA | **2.42 ± NA** | 9.03 ± NA | 9.30 ± NA | **8.93 ± NA** |

## C EVALUATION UNDER DATASET SHIFT

Dataset shift is a challenging problem in which the dataset composition and/or distribution changes over time. This scenario poses significant challenges to the application of machine learning algorithms, because these models tend to assume similar distributions between their train and test sets. Dataset shift scenarios are of particular interest to uncertainty modeling (Ovadia et al., 2019), as it enables researchers to evaluate algorithms' robustness and adaptability. We now evaluate PIVEN's ability to adjust to this challenging scenario.

For our evaluation, we chose the Flight Delays dataset (Hensman et al., 2013), which is known to contain dataset shift. We train PIVEN and our baselines on the first 700K data points and test on the next 100K test points at 5 different starting points: 700K, 2M (million), 3M, 4M and 5M. This experimental setting *fully replicates* (both in terms of dataset splits and the neural architectures used in the experiments) the one presented in He et al. (2020). The dataset is ordered chronologically throughout the year 2008, and it is evident that the dataset shift increases over time. Our experimental results – using the PICP, MPIW and RMSE metrics – are presented in Figure 7. As in our other experiments, we present the results for a confidence level of 95% (i.e. $\alpha = 0.05$).

The results show that PIVEN outperforms the baselines in MPIW metric (a result consistent with our other experiments) while achieving the desried coverage rate. Interestingly, our approach is also competitive in terms of the RMSE metric, although the value fluctuations make it difficult to reach a clear conclusion. Neither approach reached the desired PICP level in the first test set, but all were able to do so in consequent experiments. DE was able to achieve slightly higher PICP rates than the other two methods, but its PI width were larger than the other two methods. In contrast, PIVEN manages to

achieve the desired PICP levels while producing smaller intervals. It should be noted that the dataset exhibits seasonal effect between 2M and 3M samples, which causes variance in performance. This phenomenon is described in He et al. (2020). We hypothesize that the reason PIVEN outperforms DE in terms of PI width is due to the fact that the former aims to optimize MPIW, whereas DE aims to optimize for negative log-likelihood (Lakshminarayanan et al., 2017). Additionally, the Flight Delays dataset is known to have a non-Gaussian distribution (Novianingsih & Hadianti, 2014) which invalidates the basic assumption applied by DE.

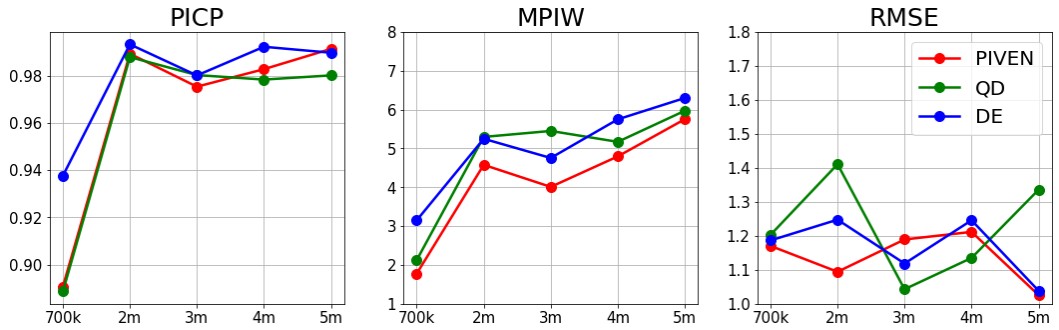

Figure 7: Compassion between PIVEN and the baselines in terms of PICP (left), MPIW (center) and RMSE (right) for $\alpha = 0.05$.

## D    FULL RESULTS - PI PERFORMANCE AS A FUNCTION OF COVERAGE

We argue that QD only focuses on the fraction $c$ of the training set which was successfully captured by the PI. In this experiment we study the effect of changes in the coverage levels (represented by $\alpha$) on the width of the PI produced both by PIVEN and QD. Additionally, we examine the affect of $\alpha$ on the value prediction. In Figures 8, 9 and 10, we present our results – measured by PICP, MPIW and RMSE – on all UCI datasets. As expected, PIVEN consistently outperforms QD in the above metrics.

**PI Coverage.**   Both methods are able to achieve the coverage desired by $\alpha$, which decreases as $\alpha$ increases, as expected.

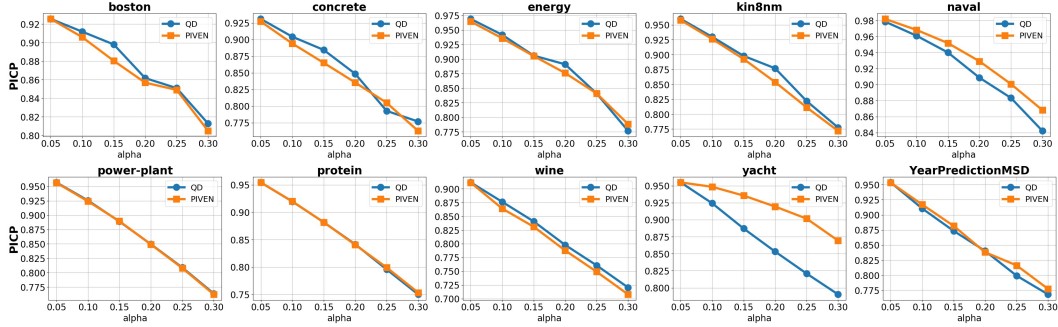

Figure 8: comparing PICP between QD and PIVEN over different values of alpha.

**PI Width.** QD's inability to consider points outside the PI clearly degrades its performance. PIVEN, on the other hand, is able to reduce MPIW consistently over all datasets.

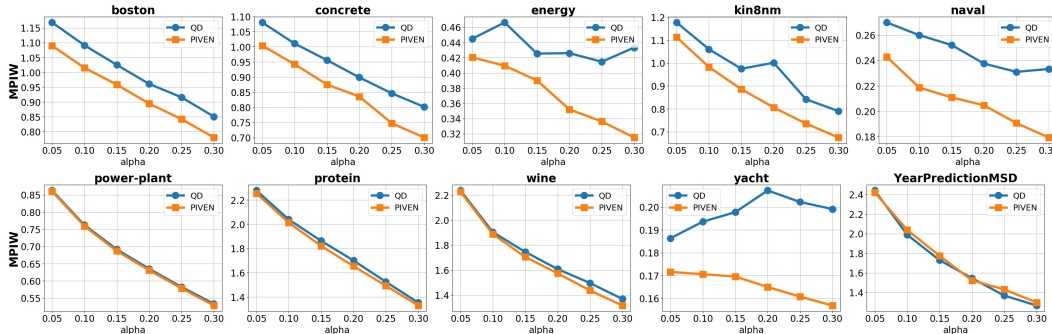

Figure 9: comparing MPIW between QD and PIVEN over different values of alpha.

**Value prediction accuracy.** Due to the way QD performs its value prediction (i.e., middle of the PI), the value prediction is not able to improve, and in fact degrades. Contrarily, PIVEN's robustness enables it to generally maintain (with a slight decrease) its performance levels.

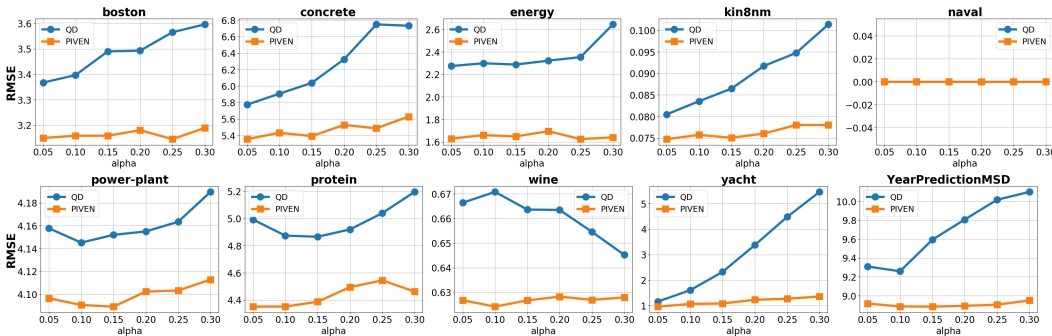

Figure 10: comparing RMSE between QD and PIVEN over different values of alpha.

## E    IMDB AGE ESTIMATION TRAINING PROCESS AND ROBUSTNESS TO OUTLIERS

### E.1    TRAINING PROCESS

In the following figures we present comparisons of the training progression for PIVEN, QD and NN on the MAE, PICP and MPIW evaluation metrics. We used 80% of images as the training set while the remaining 20% were used as the validation set (we did not define a test set as we were only interested in analyzing the progression of the training). For the MAE metric, presented in Figure 11, we observe that the values for QD did not improve. This is to be expected since QD does not consider this goal in its training process (i.e., loss function). This result further strengthens our argument that selecting the middle of the interval is often a sub-optimal strategy for value prediction. For the remaining two approaches – NN and PIVEN– we note that NN suffers from overfitting, given that the validation error is greater than training error after convergence. This phenomena does not happen in PIVEN which indicates robustness, a result which further supports our conclusions regarding the method's robustness.

For the MPIW metric (Figures 12), PIVEN presents better performance both for the validation and train sets compared to QD. Moreover, we observe a smaller gap between the error produced by PIVEN for the two sets – validation and training – which indicates that PIVEN enjoys greater robustness and an ability to not overfit to a subset of the data. Our analysis also shows that for the PICP metric (Figure 13), PIVEN achieves higher levels of coverage.

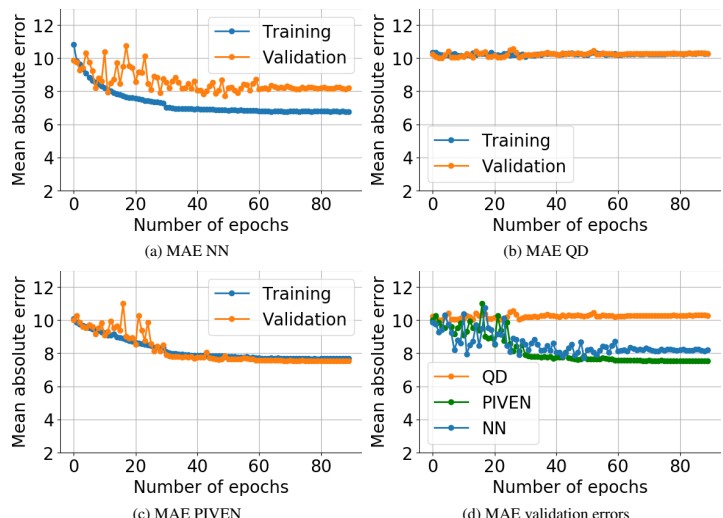

Figure 11: Comparison of MAE metric in the training process. We observe that the values for QD (b) do not improve, which is expected since QD does not consider value prediction in its loss function. Moreover, we note that NN suffers from overfitting, given that the validation error is greater than the training error after convergence. This phenomena do not affect PIVEN, thus providing an indication of its robustness.

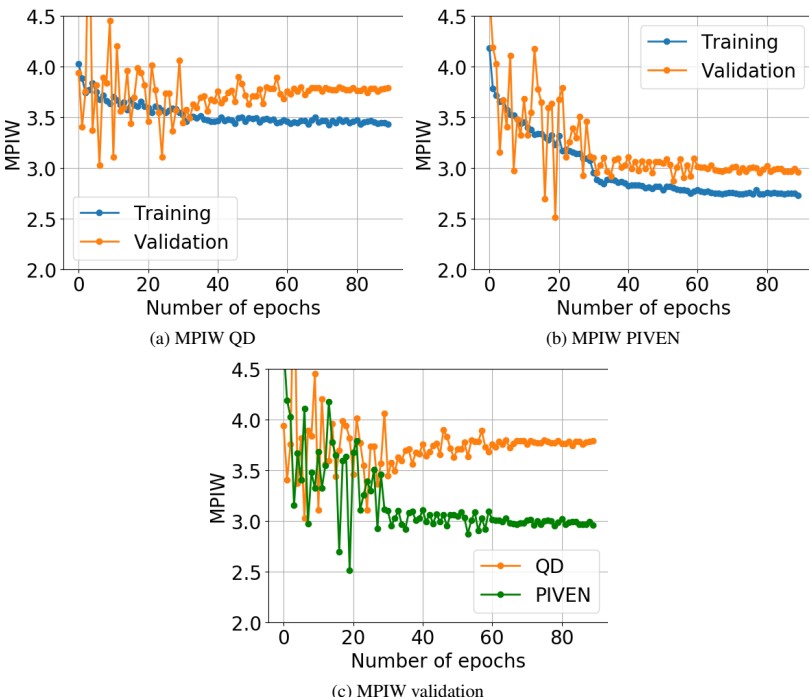

Figure 12: Comparison of the MPIW metric between QD and PIVEN in the training process. As can be seen, PIVEN significantly improves over QD, and has a smaller gap between training and validation errors.

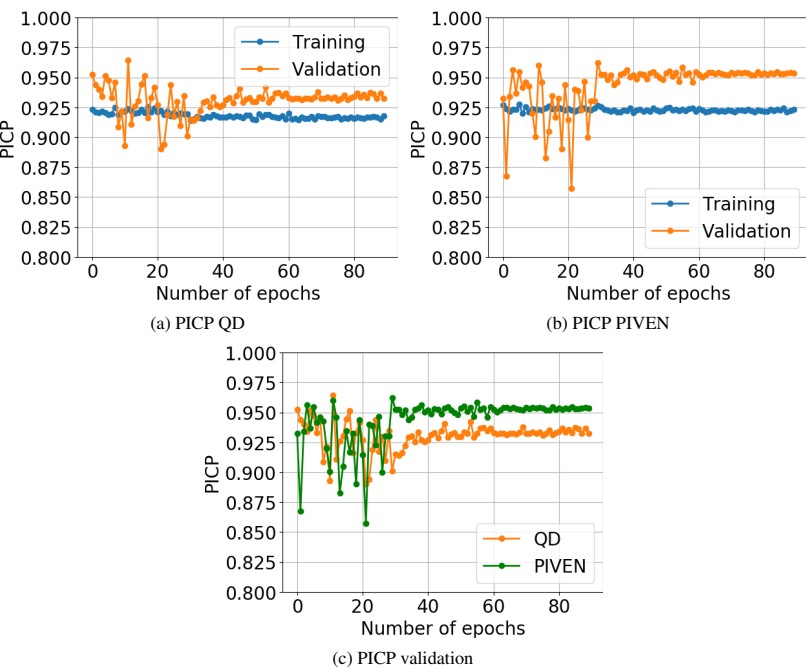

Figure 13: Comparison of PICP metric between QD and PIVEN in the training process. PIVEN achieves higher coverage when two methods converges.

### E.2 ROBUSTNESS TO OUTLIERS

Since PIVEN is capable of learning from the entire dataset while QD learns only from data points which were captured by the PI, it is reasonable to expect that the former will outperform the latter when coping with outliers. In the IMDB age estimation dataset, we can consider images with very high or very low age as outliers. Our analysis shows that for this subset of cases, there is a large gap in performance between PIVEN and QD. In Figure 14 we provide several images of very young/old individuals and the results returned by the two methods. We can observe that PIVEN copes with these outliers significantly better.

True age: 75.0Y
Predicted age QD: 37.9Y
Uncertainty: 12.9Y-62.9Y

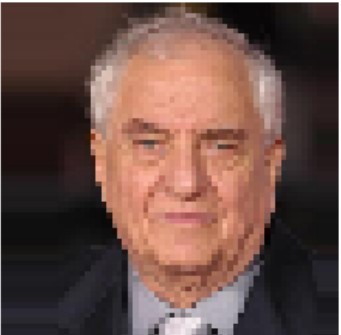

True age: 75.0Y
Predicted age PIVEN: 78.2Y
Uncertainty: 72.0Y-82.8Y

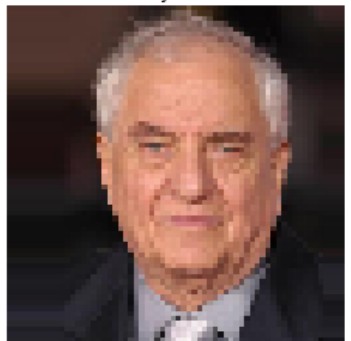

True age: 14.0Y
Predicted age QD: 36.4Y
Uncertainty: 15.2Y-57.6Y

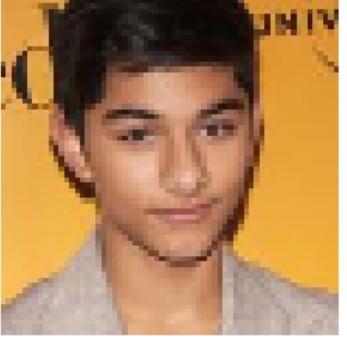

True age: 14.0Y
Predicted age PIVEN: 17.6Y
Uncertainty: 9.1Y-26.9Y

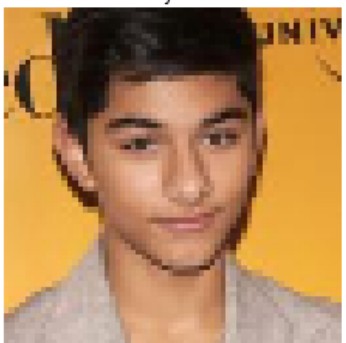

True age: 68.0Y
Predicted age QD: 37.9Y
Uncertainty: 12.0Y-63.8Y

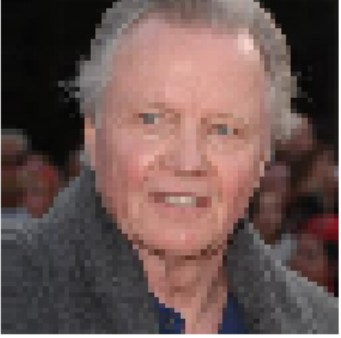

True age: 68.0Y
Predicted age PIVEN: 75.3Y
Uncertainty: 64.8Y-83.3Y

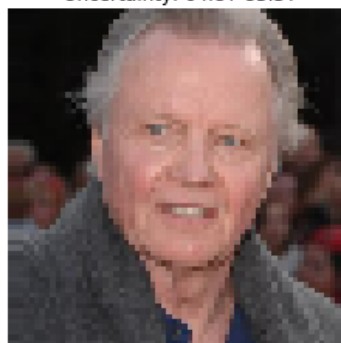

Figure 14: The predictions produced for outliers (i.e., very young/old individuals) by both PIVEN and QD for the IMDB age estimation dataset. The results for QD are on the left, results for PIVEN on the right.

