# OpenReview forum: "PIVEN: A Deep Neural Network for Prediction Intervals with Specific Value Prediction"
_ICLR.cc/2021/Conference — Reject_

### Official Review · AnonReviewer1 · 2020-10-23
**PIVEN**

**Rating:** 6
**Confidence:** 4

**Review:**

This paper proposes a new objective function for training regression networks with prediction intervals. The goal is to provide tight confidence bounds to accompany predictions, which is of course important for practical deployments of ML systems where uncertainty quantification is critical. Previous work has largely assumed that uncertainty is symmetrical, or even Gaussian distributed, which is often not the case in practice. The innovation of this work is to simultaneously predict the bounds for a given confidence level and a point prediction within those bounds, which is not necessarily the mean. It is accomplished by predicting 3 values: an upper bound, a lower bound, and a mixing parameter that allows making the point estimate as a weighted sum of the bounds. Experimental results are provided on 3 datasets, with 2 baseline methods compared against.

The strong points of this paper are:
+ The problem being addressed is important.
+ The approach is relatively simple.
+ The paper is clear and well-written.
+ The metrics introduced for evaluation make sense.
+ The results are strong and improve over the baselines.
+ The choice of how to produce the point prediction is well motivated and backed up by ablation experiments.

Weak points:
- Since the method produces bounds rather than a distribution, it's necessary to know at training time what confidence levels you are interested in. It also therefore can't be evaluated on standard negative log likelihood metrics, or produce something akin to accuracy vs rejection-fraction plots. This is potentially a big problem for practical deployment, since the tradeoffs of alpha vs RMSE can't be evaluated without retraining, and the same model can't be used for multiple downstream tasks with different confidence requirements.
- The claims about using PIVEN in an ensemble context don't seem to be experimentally validated anywhere.
- The NN baseline for large-scale image datasets seems unnecessarily weak. Why not predict both mean and variance (basically DE, but a single model)? That way you could at least compute PICP and MPIW.
- In table 1 it's not at all clear what the bold numbers are. They don't seem to correspond to the best result in each row.
- In table 2, why is PICP not provided? It seems like the value should not be identical between the different experiment arms.
- It would have been nice to see an evaluation under dataset shift, similar to the "Flight Delays" experiment in https://arxiv.org/pdf/2007.05864.pdf

Overall, this seems like a simple and valuable technique that addresses an important problem, but its applicability is somewhat limited by the fact that it produces bounds rather than a distribution.

---

> ### Author Response · Authors · 2020-11-14
> **Response to Review #1 (1 out of 4)**
>
> Thank you for your constructive comments. Please see our responses to each of your concerns below:
>
> **1. “Since the method produces bounds rather than a distribution, it's necessary to know at training time what confidence levels you are interested in. It also therefore can't be evaluated on standard negative log likelihood metrics, or produce something akin to accuracy vs rejection-fraction plots. This is potentially a big problem for practical deployment, since the tradeoffs of alpha vs RMSE can't be evaluated without retraining, and the same model can't be used for multiple downstream tasks with different confidence requirements."**
>
> We thank the reviewer for this insightful comment. Indeed, one would have to know  what are the desired confidence levels prior to training the model. This shortcoming is shared by all methods in the field (including our baselines), and addressing it is a topic for future work. We hypothesize that a DRL-based approach that weights various performance metrics based on a user-provided preference could be used to address this problem in an elegant manner.
>
> **2. “The claims about using PIVEN in an ensemble context don't seem to be experimentally validated anywhere."**
>
> We thank the reviewer for bringing the need to clarify this point to our attention. PIVEN used an ensemble of models for the UCI datasets in order to be consistent with previous work and the reported results of our baselines. We have updated the paper (Section 5.3) to clarify this point.
>
> **3. “The NN baseline for large-scale image datasets seems unnecessarily weak. Why not predict both mean and variance (basically DE, but a single model)? That way you could at least compute PICP and MPIW."**
>
> We thank the reviewer for this insightful suggestion. We have conducted an additional set of experiments in which we test the Deep Ensemble baseline with one ensemble, i.e. only outputs the mean variance. The results are reported in Table 3 in Section 5.5.
>
> **4. “In table 1 it's not at all clear what the bold numbers are. They don't seem to correspond to the best result in each row."**
>
> Our experimental setting strictly follows the one reported in QD, which is also one of our baselines. For results to be in bold, the algorithm must reach the desired PICP level. For the first two datasets, no algorithm reached the desired result. We have updated the paper to clearly state this fact in Section 5.4:
>
> “Results on regression benchmark UCI datasets comparing PICP and MPIW. Best performance defined as in Pearce et al. (2018): every approach with PICP≥0.95 was defined as best for PICP. ForMPIW, best performance was awarded to lowest value. If PICP≥0.95 for neither, the largest PICP was best, and MPIW was only assessed if the one with larger PICP also had smallest MPIW.”
>
> **5. “In table 2, why is PICP not provided? It seems like the value should not be identical between the different experiment arms."**
>
> The subject of the PICP is of course important, but is adequately addressed in the paper. We refer the reader to the following text  in Section 5.4: “the PICP values are identical for all variants and are therefore omitted—see full results in Appendix B”
>
> **6. “It would have been nice to see an evaluation under dataset shift, similar to the "Flight Delays" experiment in https://arxiv.org/pdf/2007.05864.pdf"**
>
> We thank the reviewer for this useful suggestion. We agree that an evaluation under dataset shift, could provide an important insight into PIVEN, and are currently running this experiment. We will report these results in the coming days. In the interest of providing the reviewer with as much time as possible to review our responses, we upload our answer to the remaining questions here.
>
> If the reviewer has any additional questions, we would be  happy to provide more clarification and revise our paper accordingly.

---

> ### Author Response · Authors · 2020-11-18
> **Evaluation of PIVEN under dataset shift**
>
> Dear Reviewer #1,
> We have evaluated PIVEN and the other baselines on the Flight Delay dataset. The results of the evaluation and relevant analysis are presented in Appendix C of the new version of our paper. We would like to stress that we have followed the experimental setting presented in [1] - both in terms of experimental setup and the architectures used in the experiment.
> The results of our evaluation show that PIVEN outperforms the two baseline in terms of MPIW (i.e., interval width), while achieving comparable results in terms of PICP and RMSE. We refer the reviewer to our paper for the full analysis.
> We would to thank again the reviewer for proposing this dataset, which enabled us to test PIVEN's effectiveness in a challenging setting.
>
> [1] Bayesian Deep Ensembles via the
> Neural Tangent Kernel - https://arxiv.org/pdf/2007.05864.pdf

---

### Official Review · AnonReviewer2 · 2020-10-29
**limited novelty and evaluation**

**Rating:** 4
**Confidence:** 4

**Review:**

Summary:
The submission considers the continuous real-valued regression problems and how to obtain accurate point predictions (specific value prediction in the text) and prediction intervals (the uncertainty of the predictions, given by [lower bound LB, upper bound UB]). The paper proposes a loss function which is the weighted combination of (i) the coverage constraint [proportion of outputs that fall between LB and UB] so that the coverage meets a required level, (ii) the prediction interval width [for those that satisfy (i), their prediction intervals should be tight], and (iii) the prediction loss [which makes sure the predictions match the outputs]. The LB, UB and predictions are parameterised using three separate heads: one for LB, one for UB and one for the prediction weight (which makes sure the prediction stays in between LB and UB. The proposed loss function and parameterisation architecture are evaluated on UCI regression datasets and two age prediction problems using image inputs.

Assessment:
Whilst I think the submission tackles an important problem that could be of interest to the ICLR community, the novelty and experimental evaluation are limited and thus I do not recommend acceptance. Below are some of my concerns/questions and I appreciate the response from the authors.

1. The key contribution is the paper is the combination of the prediction interval loss and the prediction loss. Each of these losses are not new, for example: the prediction interval loss has been considered by Pearce et al (2018). The UCI regression also do not show strong evidence that the proposed method is better than QD, in contrast to the claim of “state-of-the-art results in uncertainty modeling by the use of PIs”.

2. The use of a separate head for v is also not strongly supported by the UCI results, compared to directly parameterising the prediction (POO) or MOI. In fact, PIVEN is probably better because of the hyperparameter search over \beta.

3. The contribution of the paper includes the output head architecture and the loss function, so I’d not call this a deep neural network.

4. In several places, the paper alludes to skewed output distributions and how PIVEN can handle this better than alternatives. However, it is not clear from the experiments that this is the case. PIVEN is very poor at predicting the Sine example in the appendix -- could you clarify this? Could the "skewed distribution" argument be formalised theoretically? like the PIVEN objective is inspired from another likelihood which can support skewed outputs compared to the usual Gaussian likelihood/L2 loss?

5. it would be good to compare to the post-hoc calibration procedure for regression by Kuleshov et al (2018)

6. Does the ensemble of PIs work in practice?

minor: abstract: read-world -> real-world

---

> ### Author Response · Authors · 2020-11-14
> **Response to Review #2 (2 out of 4)**
>
> Thank you for your constructive comments. Please see our responses to your questions below:
>
> **A1:**
> While we agree that neither component is new in itself, we combine the two tasks -- PI generation and value prediction -- in a novel way that enables us to outperform the current state-of-the-art approach - QD - in the task of PI generation while achieving comparable results to DE in terms of value prediction. Moreover, the design of the auxiliary head - outputting a fraction in between the upper and lower bounds means we use a non-Gaussian observation model which is different from many of the previous approaches.
>
> An additional novelty of our approach is its ability to take into account all the points of the dataset, and not only those captured with their respective PIs. This trait enables our approach to better generalize compared to baselines such as QD, and ultimately leads to better performance.
>
> Finally, while the average improvement offered by PIVEN in Table 1 is small (4.5%), it is by no means insignificant. Moreover, one should bear in mind that for this value of alpha, 95% of samples are contained within the interval, so performance is relatively high across the board. When using smaller confidence levels, PIVEN’s relative performance is higher overall (up to 11.2%) and reaches much values for individual datasets. This is the case because of its ability to consider all points and not only those that are captured by the PI. We refer the reviewer to Table 4 for the results of this analysis.
>
> **2.1 “The use of a separate head for v is also not strongly supported by the UCI results, compared to directly parameterising the prediction (POO) or MOI”**
>
> Table 2 presents our ablation testing, where we compare partial versions of PIVEN to the full approach. The reviewer is correct that in terms of MPIW the differences are minor, but that is to be expected given the fact that the difference between the full and ablated version lies in the former’s ability to utilize out-of-PI data points to output better value predictions. Table 2 clearly shows that with regard to RMSE,  PIVEN achieves top results for all evaluated UCI datasets for value prediction.
>
> **2.2 “In fact, PIVEN is probably better because of the hyperparameter search over $\beta$.”**
>
> We did not perform any hyperparameter search over $\beta$. In fact, $\beta$ value is always set to 0.5 in all our experiments. We have updated the paper to explicitly state this fact.
>
> **4.1 “...the paper alludes to skewed output... However, it is not clear from the experiments that this is the case.”**
>
> Our experiments provide a clear empirical demonstration that we are able to model non-Gaussian distribution (see in Section 6.1). This ability is due to PIVEN’s ability to output any value prediction within the PI.
>
> **4.2 “PIVEN is very poor at predicting the Sine example in the appendix -- could you clarify this?”**
>
> We believe there may have been a misunderstanding on the side of the reviewer. The results of the Sine dataset, presented in Figure 5, clearly show that PIVEN outperforms QD. While both methods output relatively similar PIs, the value prediction produced by PIVEN closely follows the Sine distribution while QD’s value prediction remains constant.
>
> **4.3 “Could the "skewed distribution" argument be formalised theoretically? like the PIVEN objective … Gaussian likelihood/L2 loss?"**
>
> PIVEN’s objective is motivated by the calculus of variables using analytic geometry. In analytic geometry, one can find a point within an interval given a weight assigned to one of the ends. In our adaptation, PIVEN learns this weight, which is denoted as $v$. We added a formalization to the "skewed distribution" in the paper.
>
> **A5:**
> The comparison proposed by the reviewer is interesting. However, the approach proposed by Kuleshov et al is different from PIVEN and all the baselines in several important aspects. For one, Kuleshov et al propose a post-modelling step, which means that it is a model-agnostic approach, while PIVEN and the baselines are model-aware. The former treats the neural net as a black-box and attempts a post-hoc optimization, while PIVEN and the baselines are trained together with the neural net. These differences are well explained by the following quote “...a post-modelling step, which means that it is model-agnostic”
>
> Because of these differences, a fair comparison between PIVEN and Kuleshov et al cannot be carried out. In fact, the method presented in the paper proposed by the reviewer could be implemented on top of PIVEN. In the words of the authors of Kuleshov et al: “In summary, we introduce a simple technique for recalibrating the output of any regression algorithm...”
>
> **A6:**
> We use the ensemble approach in all our experiments on UCI datasets. We do so in order to ensure that our experimental setting is identical to those of our baselines, which also use ensembles. We have clarified this point in the updated version of our paper in Section 5.3.

---

### Official Review · AnonReviewer4 · 2020-10-29
**A new approach for combining prediction and uncertainty quantification in DNNs - good empirical results, explanation can be improved**

**Rating:** 7
**Confidence:** 4

**Review:**

**Quality and Clarity**

While the overall message of the paper is clear, the explanation of the method in Section 4 is a bit hard to follow. Specifically it is a bit hard to keep a track of the meaning of the multiple terms in the loss function and the associated hyper parameters (see Queries and Suggestions below).

**Originality and Significance**

The authors seek to jointly address the problem of making accurate predictions and generating tight prediction intervals by designing a new loss function that combines these two goals. I believe this work can be quite impactful as there are multiple areas where good quality prediction intervals and accurate value prediction are equally necessary.

**Strengths**

The proposed approach appears principled and seems to give good empirical results in the experiments considered.

**Weaknesses**

There seem to be multiple hyperparameters ($\lambda$,$\beta$) whose choice might affect the performance of the approach. Moreover as this seems to be the first time such a loss function is used in this context there is no evidence on good choices of hyper parameters (other than the values used in the experiments herein). I would highly appreciate an ablation study or some comments on good values of hyper parameters which can guide readers intending to apply the proposed approach to a new dataset.

**Queries and Suggestions**

1. I'm curious about the performance of the ensemble of PIVEN architectures approach. It does not seem to have been used in any of the experiments which is fine since a single model seems to be doing well. But have you tried it on any dataset? What are the settings where the improvement with an ensemble would be significant enough to compensate for the added cost?

2. Why are the MPIW results for DE on Boston and Concrete not in bold since DE seems to be the best in terms of MPIW on the two datasets?

3. Why is only a single dense layer (and not an ensemble) used when applying DE to large scale image datasets? If the backbone is pre-trained then training an ensemble of dense layers should not be too expensive. The current comparison does not seem fair since the NN baseline only has a MSE loss while the PIVEN approach uses multiple regularizers (for good MPIW, PICP etc.)

4. What is the difference between the MOI variant of PIVEN and the QD baseline?

In addition to responding to the above queries, I would recommend improving the explanation (one suggestion is to write the entire loss i.e. $\mathcal{L}_{\text{PIVEN}}$ at the beginning of Section 4 right after Figure 1 and then explain the meaning of each term in the loss, as opposed to the current approach where the terms are introduced first and then the loss is given) and commenting on good choices of the hyper parameters or including an ablation study for the same.

**Comments after Author Response**

I thank the authors for their response. Queries 1,2, and 4 have been adequately addressed. Regarding Query 3, I appreciate the addition of the Deep Ensemble results though I find that the text of Section 5.5 has not been changed to reflect the same. Specifically the paper still says "For the IMDB age prediction dataset, results show that PIVEN outperforms both baselines across all metrics". This is now incorrect since there is a third baseline, DE, which appears to outperform/match PIVEN for this dataset. However the explanation that this is because the population age is approximately Gaussian makes sense to me and so keeping in mind the good performance of PIVEN on the other datasets, and the improved explanation and added ablation study for hyper parameters, I recommend accepting the paper as long as the relevant corrections are made in Section 5.5.

---

> ### Author Response · Authors · 2020-11-15
> **Response to Review #4 (4 out of 4)**
>
> Thank you for your constructive comments. Please see our responses to each of your concerns below:
>
> **A1:** We thank the reviewer for bringing the need to clarify this point to our attention. PIVEN used an ensemble of models for the UCI datasets in order to be consistent with previous work and the reported results of our baselines.  For all other experiments (large and synthetic datasets), both PIVEN and the evaluated baselines used a single model. We add a clarification to this point in the paper in Section 5.3.
>
> **A2:** Our experimental setting strictly follows the one reported in QD, which is also one of our baselines. For results to be in bold, the algorithm must reach the desired PICP level. For these two datasets, no algorithm reached the desired result. We have updated the paper to clearly state this fact in Section 5.4:
>
> **3.1 “Why is only a single dense layer ... training an ensemble of dense layers should not be too expensive”**
>
> We thank the author for highlighting this point. In pre-trained we meant that the final layer (i.e. PIVEN) was added to architectures whose weights were not randomly initialized. Based on recommendations in Kaggle, for the RSNA bone age dataset we began with an architecture that was pre-trained on ImageNet and then conducted additional training on the dataset itself. For IMDB, we followed the experimental setting performed in previous studies [1] and trained our own architecture. Because of the need to run a 5-fold evaluation, multiple runs would have been computationally prohibitive. We have updated our paper to further clarify these points.
>
> **3.2 “The current comparison does not seem fair ...  PIVEN approach uses multiple regularizers (for good MPIW, PICP etc.)”**
>
> We thank the reviewer for this insightful suggestion. We have conducted an additional set of experiments in which we test the Deep Ensemble baseline with one ensemble, i.e. only outputs the mean variance. The results are reported in Table 3 in Section 5.5.
>
> **A4:** QD loss optimizes only for the PI (L, U) without optimizing the value prediction.
> In MOI, we added the optimization for the point prediction,  which is a variant of PIVEN where $v=0.5$.
> In other words, MOI loss is  $\mathcal{L}_{PI} + \ell(0.5L + 0.5U, y_t)$ where $\ell$ is the regression loss.
> We add a clarification to this point in the paper:
>
> “While both return the middle of the PI as point prediction, MOI differs from QD by an additional component in its loss function”
>
> **5. “... I would recommend improving the explanation ... and commenting on good choices of the hyper parameters or including an ablation study for the same.”**
>
> We thank the reviewer for this highly insightful comment. First, we have updated Section 4.1 along the lines proposed by the reviewer, The section now contains an overview of our loss function and the rationale behind its implementation.
>
> Secondly, we have conducted two additional sets of experiments to explore the effect of changes in the $\lambda$ and $\beta$ parameters used in our model. The new results appear in the updated of our paper in Section 6.3. As shown in the results, $\beta$ balances our goals of creating narrow PIs and producing accurate value predictions. Different values result in very different results with respect to these two measures. $\lambda$ balances our goals of capturing as many points as possible within our PIs while producing tight PIs.
>
> We believe that this new analysis will enable practitioners to better adapt our approach to their individual datasets and preferences in terms of coverage and tightness of PI bounds. We thank the reviewer for this useful suggestion.
>
> **6. “there seem to be multiple hyperparameters (λ,β) whose choice might affect the performance of the approach.”**
>
> $\lambda$ was set to 15 (as done in QD) and $\beta$ was set to 0.5 in all our experiments. We have also added a comprehensive analysis of the effect these parameters’ values have on performance (please see our response to question #5).
>
> **7.  “Moreover as this seems to be the first time such a loss function is used in this context there is no evidence on good choices of hyper parameters (other than the values used in the experiments herein).”**
>
> All hyperparameters used in our evaluation were identical to those used by the QD baseline.  This point is explicitly stated in our paper:
>
> “To ensure that our comparison with the state-of-the-art baselines is accurate, we first set the parameters of our neural nets so that they produce the results reported in Pearce et al.(2018).”
>
> **8.  “I would highly appreciate an ablation study or some comments on good values of hyper parameters which can guide readers intending to apply the proposed approach to a new dataset.”**
>
> Please see our response to question #5.
>
>
> If the reviewer has any additional questions, we would be happy to provide more clarification.
>
> [1] Yang, Tsun-Yi, et al. "SSR-Net: A Compact Soft Stagewise Regression Network for Age Estimation." IJCAI, 2018.

---

### Official Review · AnonReviewer3 · 2020-11-02

**Rating:** 6
**Confidence:** 3

**Review:**

The authors propose a method for predicting a prediction interval. The authors compare this method to deep ensembles and show improved performance on a set of benchmarks.


The are several things I like about this paper:
- The paper is written quite clear
- Uncertainty calibration is an important topic.
- A lot of benchmarks are considered
- The method is easy to implement.

These are the things  that could be improved:

1. The authors say ""First,  our approach is the first to propose an integrated architecture capable of producing both PIs and exact value predictions."

This is not correct.  A simple approach such as Lakshminarayanan et al., 2017 will produces the same, even without using ensembles.  In that work,  the predictions parameterize a Gaussian with state dependent mean and variance. From this quantiles and thus prediction intervals can be derived on top of an "exact value prediction" which would be the mean.

2.  One of the papers main weakness is the lack of distinction between epistemic and aleatoric uncertainty:


Section 2:
The way  Section 3 is formulated it speaks about aleatoric uncertainty:    P(L_1 < y_i < U_i) is the probability that y_i falls into th ebound defined by L_i and U_i and thus is "data uncertainty" (aleatoric) and not due to the lack of knowledge given by the model (epistemic).  it is thus similar to a Softmax in classification or  a neural network that predicts both mean and variance (as in  (Lakshminarayanan et al., 2017)). Indeed as Section 4 show,  the model does not contain any parametric uncertainty.


3. Results and method comparisons:

Section 4:
I am not sure why RMSE is a useful metric when proposing a method for uncertainty calibration.  Secondly, as mentioned above the comparison to deep ensembles is not fair, as deep ensembles will also model epistemic uncertainty.  I would just use an ensemble of size 1 for a more accurate comparison (thus excluding epistemic uncertainty). Thirdly, the two other metrics need to be explained (at least write them once without acronyms.)

Overall,  the improvements appear  to be only marginal improvements.

4. Related work:
How is this work different from  [1] and why wasn't it compared to this work?

[1] Salem, Tárik S., Helge Langseth, and Heri Ramampiaro. "Prediction Intervals: Split Normal Mixture from Quality-Driven Deep Ensembles." Conference on Uncertainty in Artificial Intelligence. PMLR, 2020.

---

> ### Author Response · Authors · 2020-11-15
> **Response to Review #3 (3 out of 4)**
>
> Thank you for your constructive comments. Please see our responses to each of your concerns below:
>
> **1.“This is not correct. A simple approach such as Lakshminarayanan et al. ... which would be the mean.”**
>
> All methods that produce PIs can also produce value predictions (e.g., QD, SQR). That is not the material point. All previous approaches simply take the middle of the interval as their point prediction. This approach works well when the data within the interval is distributed uniformly or as a Gaussian (as explicitly stated in the QD paper) but is less effective in other cases, as shown in our experiments in Section 6.1.
>
> Additionally, in the study mentioned by the reviewer, the authors acknowledge that they make the assumption of a Gaussian distribution: “By treating the observed value as a sample from a (heteroscedastic) Gaussian distribution with the predicted mean and variance, we minimize the negative log-likelihood criterion”. This assumption also shared with the one-ensemble approach mentioned by the reviewer.
> Based on this comment, we have updated our paper to better explain this point in Section 4.4
>
> **2. “One of the papers main weakness is the lack of distinction between epistemic and aleatoric uncertainty”**
>
> We thank the reviewer for bringing this to our attention. We have updated Section 4.3 to clarify this important point.
> It is important to note that a single PIVEN model captures data certainly while the ensemble of PIVEN models captures parametric uncertainty. This point is described in Section 4.2 of the original submitted version (“While PIs produced by a single PIVEN…”) and in Section 4.3 in the updated version (where we further clarify this point).
>
> **3. “I am not sure why RMSE is a useful metric ... for uncertainty calibration”**
>
> We do not use RMSE to measure uncertainty. As in our baselines, uncertainty is measured by the PICP and MPIW metrics. This distinction is described in Section 5.2.
>
> **4. “... the comparison to deep ensembles is not fair .. just use an ensemble of size 1”**
>
> We thank the reviewer for raising this interesting point, as it would enable us to evaluate the trade-offs between optimizing for our model’s mean-variance and epistemic uncertainty. We therefore have conducted additional experiments with an ensemble size of 1. We added the results in the Appendix (Table 6) and updated the paper (Section 5.4) to elaborate on this point.
>
> **5. “the two other metrics need to be explained (at least write them once without acronyms.)”**
>
> The metrics (full names and formulae) are presented in Section 3.
>
> **6.“Overall, the improvements appear to be only marginal improvements.”**
> The results presented in Table 1 are for $\alpha=0.05$ because this is the value reported in all leading baselines. We refer the reviewer to Table 4 in Section 6.2 for an evaluation of PIVEN’s relative performance for multiple values of alpha. For $\alpha=0.05$, PIVEN achieves a 4.5% improvement over the baselines, and we consider this improvement to be significant. Additionally, one should bear in mind that for this value of alpha, 95% of samples are contained within the interval, so performance is relatively high across the board. When using smaller confidence levels, PIVEN’s relative performance is higher overall (up to 11.2%) and reaches much values for individual datasets.
>
> **7. “Related work: How is this work different from [1] and why wasn't it compared to this work?”**
>
> We thank the reviewer for bringing this work to our attention. Since this paper was published shortly before our paper was submitted, we were not aware of its publication. We have thoroughly read the paper. The presented approach is similar to ours in the sense that it attempts to simultaneously output  both a PI and point prediction. However, there are two important differences:
>
> **a.** [1] does not express the point prediction as a function of the interval bounds. This disconnect forces the authors to add an additional component to their loss in order to bound the point prediction. This approach yields sub-optimal results compared to PIVEN (please see our reference to the experiments below).
>
> **b.** [1] actually combines two separate improvements. One is the point prediction mentioned above, and the other is the SNM approach which is not a novelty. SNM is used to boost the performance of both the approach presented in [1] and its baselines (we share the same baselines as [1]). We add a mention of this study in our paper and explain that  PIVEN’s can utilize the SNM approach for additional improvement in Section 4.3.
>
> If we set SNM aside, the difference between PIVEN and [1] centers around the way by which the point prediction is produced.
> In fact, we already compare PIVEN to this version of [1]: our analysis (Section 5.4) presents PIVEN’s performance in cases where the point prediction is not expressed as a function of the bounds. Table 2 clearly show that the approach presented in [1] yields inferior results.

---

### Author Response · Authors · 2020-11-15
**Summary of Main Revisions**

We want to thank all the reviewers for their effort in reading the paper and providing constructive comments. We have addressed all the main concerns, and have updated the paper accordingly. The main updates are as follows:

1. We added an analysis of PIVEN’s  hyperparameters and their effect on the various performance metrics used in our study.
2. We have clarified (both the reviewers and in the study itself) several points regarding the experimental setup.
3. We have added an additional baseline (DE with ensemble size=1) for all experiments.
4. We have added a new section (4.3) that discusses PIVEN’s ability to capture both aleatoric and parametric uncertainties.
5. We have added an evaluation of PIVEN under dataset shift (Appendix C).


Additionally, we have performed the following:
1. Updated our related work to address questions raised by the reviewers.
2. Provided additional information regarding skewed-normal distributions.

Finally, we have also responded to the specific comments of each reviewer in replies to their reviews.

---

### Decision · Program_Chairs · 2021-01-07
**Final Decision**

**Decision:**

Reject

**Comment:**

The paper presents PIVEN, a deep neural network that produces a prediction interval in addition to a specific point prediction.  PIVEN is distribution free and does not assume symmetric intervals.

All the reviewers agree that the paper investigates an important problem and the paper is well-written. The reviewers also identified a couple of weak points, namely:
- Novelty: The key idea seems to a combination of prediction loss (common) and prediction interval loss which has been proposed by Pearce et al. 2018.
- Claims that PIVEN outperforms existing methods (QD and DE) empirically as some of the improvements seem marginal.

Given these concerns, I think the current version falls a bit short of the acceptance threshold unfortunately. I encourage the authors to revise the draft and resubmit to a different venue.